# Visual Deprivation Alters Functional Connectivity of Neural Networks for Voice Recognition: A Resting-State fMRI Study

**DOI:** 10.3390/brainsci13040636

**Published:** 2023-04-07

**Authors:** Wenbin Pang, Wei Zhou, Yufang Ruan, Linjun Zhang, Hua Shu, Yang Zhang, Yumei Zhang

**Affiliations:** 1Department of Neurology, Beijing Tiantan Hospital, Capital Medical University, Beijing 100070, China; pangwenbin623@gmail.com; 2China National Clinical Research Center for Neurological Diseases, Beijing 100070, China; 3Beijing Key Lab of Learning and Cognition, School of Psychology, Capital Normal University, Beijing 100048, China; 4School of Communication Sciences and Disorders, Faculty of Medicine and Health Sciences, McGill University, Montréal, QC H3A 1G1, Canada; 5Centre for Research on Brain, Language and Music, Montréal, QC H3A 1G1, Canada; 6School of Chinese as a Second Language, Peking University, Beijing 100871, China; 7State Key Laboratory of Cognitive Neuroscience and Learning, Beijing Normal University, Beijing 100875, China; 8Department of Speech-Language-Hearing Sciences and Center for Neurobehavioral Development, The University of Minnesota, Minneapolis, MN 55455, USA; 9Department of Rehabilitation, Beijing Tiantan Hospital, Capital Medical University, Beijing 100070, China

**Keywords:** voice recognition, face recognition, early blind, functional connectivity

## Abstract

Humans recognize one another by identifying their voices and faces. For sighted people, the integration of voice and face signals in corresponding brain networks plays an important role in facilitating the process. However, individuals with vision loss primarily resort to voice cues to recognize a person’s identity. It remains unclear how the neural systems for voice recognition reorganize in the blind. In the present study, we collected behavioral and resting-state fMRI data from 20 early blind (5 females; mean age = 22.6 years) and 22 sighted control (7 females; mean age = 23.7 years) individuals. We aimed to investigate the alterations in the resting-state functional connectivity (FC) among the voice- and face-sensitive areas in blind subjects in comparison with controls. We found that the intranetwork connections among voice-sensitive areas, including amygdala-posterior “temporal voice areas” (TVAp), amygdala-anterior “temporal voice areas” (TVAa), and amygdala-inferior frontal gyrus (IFG) were enhanced in the early blind. The blind group also showed increased FCs of “fusiform face area” (FFA)-IFG and “occipital face area” (OFA)-IFG but decreased FCs between the face-sensitive areas (i.e., FFA and OFA) and TVAa. Moreover, the voice-recognition accuracy was positively related to the strength of TVAp-FFA in the sighted, and the strength of amygdala-FFA in the blind. These findings indicate that visual deprivation shapes functional connectivity by increasing the intranetwork connections among voice-sensitive areas while decreasing the internetwork connections between the voice- and face-sensitive areas. Moreover, the face-sensitive areas are still involved in the voice-recognition process in blind individuals through pathways such as the subcortical-occipital or occipitofrontal connections, which may benefit the visually impaired greatly during voice processing.

## 1. Introduction

Humans recognize a person’s identity primarily by their face and voice. Functional magnetic resonance imaging (fMRI) studies in human and nonhuman primates have revealed a set of cortical areas specialized for face processing [1,2,3,4,5]. Face-sensitive areas of humans are most selectively and reliably located in two regions, namely, the fusiform face area (FFA) [6,7,8], which is predominantly implicated in face-identity recognition, and the occipital face area (OFA), which is primarily involved in sensory information representation [5,9,10]. In analogy to face processing, voice-sensitive regions are localized in the superior temporal gyrus/sulcus (STG/STS), particularly on the right side [11,12,13] with the anterior STG/STS, which is closely related to voice identity perception, and the posterior STG/STS, which is more involved in the processing of acoustic properties of voices [14,15,16]. In addition, extratemporal regions, including the amygdala and prefrontal regions, are also involved in voice recognition [15,17,18].

In a phone call, the moment we recognize an acquaintance by the voice, we can also recall the face. There is accumulating evidence supporting the crossmodal interaction of facial and vocal information during speaker recognition [19,20,21,22]. Specifically, there are direct functional and structural links between the temporal voice areas (TVA) and FFA [23,24,25]. These findings suggest that our brain integrates multisensory information to recognize a person. The two most important mechanisms for determining personal identity, i.e., voice and face recognition, are not isolated from each other [26,27,28,29]. Some researchers emphasized the similarities in the cognitive and neural mechanisms underlying face and voice perception (for a review, see [28]) based on the “metamodal” principle of brain organization [30]. According to the “metamodal” principle, the human brain is organized based on assigned functions or computations regardless of sensory input modality [31,32]. The distinct voice and face perception systems, therefore, may share similar computational mechanisms in support of identity recognition across auditory and visual modalities.

People who lost vision in their early life provide researchers with an exceptional opportunity to investigate the “metamodal” hypothesis for identity recognition. For instance, an fMRI study in congenitally blind individuals observed increased activation in STS and FFA for the “vocal versus nonvocal” condition when compared with sighted controls [33]. Enhanced activation in FFA was also found specifically as early blind participants responded to normal voices relative to scrambled voices [34], and congenitally blind participants responded to person–voice incongruent stimuli relative to person–voice congruent stimuli [35]. Neuroimaging studies on affective information perception found that emotions conveyed by voices could be decoded in the face-sensitive areas including right FFA in the blind [36]. In addition, increased right amygdala activation to emotional voices was found in congenitally blind individuals compared with sighted participants [37]. These task-based neuroimaging findings provide substantial evidence for the plastic changes that occurred in both the voice and face systems during voice processing induced by vision loss, indicating that the involvement of FFA in identity recognition does not necessarily rely on visual input.

Several interesting issues concerning the reorganization of neural mechanisms of voice recognition in the blind remain to be further clarified. For instance, does visual deprivation reshape the internal links within the intact voice system and disrupted face system? More importantly, do the interactions between the voice and face systems retain or cease to exist in blind individuals? Lifelong blindness presumably involves behavioral adaptations, which may lead to enhanced auditory memory and attention. Several resting-state fMRI (rs-fMRI) studies have demonstrated extensive alterations of brain functional connectivity (FC) after visual deprivation [38], including the generally decreased FCs between the occipital visual cortices and temporal multisensory cortices and increased FCs between visual cortex and regions important for memory and cognitive control of attention [39,40,41]. However, as human voice processing depends on a distributed network of interlinked voice-sensitive areas [17], there may be different patterns of changes in the pathways between the anatomically separable, functionally specialized voice-sensitive areas (e.g., the anterior/posterior TVA and amygdala) and face-sensitive areas (e.g., FFA and OFA). Furthermore, as blind individuals have lost the ability to process visual input, including facial information, they almost exclusively rely on voices to identify others. Several studies using a “training-recognition” paradigm have revealed compensatory enhancements for voice recognition in congenitally blind individuals. These individuals learn faster in voice-recognition training, recognize learned speakers with higher accuracy, and respond faster than their sighted counterparts [35,42]. Their superior performance in voice recognition persists even two weeks after training [43]. However, it remains unclear whether the heightened voice-recognition ability in blind individuals is due to altered functional connectivity (FC) in the neural substrates for voice processing.

To investigate the plastic changes in the FC patterns of the voice perception network in blind compared to sighted participants, the present study quantitatively evaluated the internal FCs of different subareas involved in voice processing, the FCs between voice-sensitive and face-sensitive areas, and whether the FC changes among these areas could predict the superior voice-recognition ability in early blind individuals. In light of our recent study revealing a strong language familiarity effect in voice recognition in blind individuals [44], we included both Chinese and Japanese materials to verify the significant effect of recognizing voices spoken in a nonnative language. Findings from this study will provide insights into how visual deprivation affects the voice and face neural systems that process identity information in different sensory modalities and how the affected systems cooperate functionally during speaker recognition.

## 2. Materials and Methods

### 2.1. Participants

Twenty early blind adults (EB; 5 females; mean age = 22.6 years; age range: 18–35 years) were recruited from the Special Education College at Beijing Union University and local communities in Beijing. All the blind participants had complete vision loss or no more than rudimentary sensitivity for brightness differences with no pattern vision. Four had become completely blind no later than the age of four and the others were congenitally blind (see Appendix A for a full description). We also recruited 22 sighted control participants (SC; 7 females; mean age = 23.7 years; age range: 20–38 years) who were matched to the blind participants for age, educational level, and musical experience. All participants were right-handed according to Edinburgh Handedness Inventory [45]. All the blind and sighted participants reported normal hearing and no history of neurological or psychiatric disorders. All participants were native Chinese speakers with no prior experience of Japanese. The study was approved by the Institutional Review Board of Peking University (approval code: #2015-12-06). Each participant provided written informed consent to their participation in the experiment.

### 2.2. Stimuli and Procedure of the Behavioral Experiment

Stimulus material for the voice-recognition task consisted of 15 Mandarin Chinese sentences and 15 Japanese sentences, which were selected from a corpus used in our previous study [46]. The number of syllables across sentences in both languages was kept at 15 on average. The average duration of sentences was 2695 ms (SD = 55 ms) for Chinese (CN) and 2676 ms (SD = 55 ms) for Japanese (JP). Sentences were read naturally by five female native speakers of each language, resulting in a total of 150 stimuli. All the sentences were perceived as having no discernible idiosyncratic talker characteristics (e.g., unusual phonetic or prosodic properties such as creaky voice). The 16-bit digital audio recordings were sampled at 44.1 kHz. The stimulus materials were volume balanced using Praat software (http://www.fon.hum.uva.nl/praat/, accessed on 2 February 2017) and were presented over headphones.

We adopted a well-established paradigm [42,43,47,48] for the speaker recognition experiment to assess the voice-recognition ability of our participants. Each of the blind and sighted participants performed the speaker recognition experiment in both language conditions (CN and JP) and the order of language was counterbalanced across participants. In each language condition, the experiment consisted of four sessions: familiarization phase, practice phase, generalization phase (GP), and delayed memory phase (DP) (Figure 1). The tests in the first three phases were conducted in order on the same day and the delayed memory phase was set after two weeks. Each participant was tested individually in a quiet room.

Familiarization phase

The familiarization phase was introduced first to help participants associate the speakers with the corresponding voices. In each trial, each participant heard a number designating the speaker (i.e., No. 1–5) followed by 1 of the 5 training sentences read by that speaker. Trials were blocked by sentences. Each sentence was read by all five speakers with two repetitions. Thus, each participant heard 5 sentences × 5 speakers × 2 times = 50 trials in total.

Practice phase

After familiarization, participants were trained to identify the voice of each speaker. The sentence stimuli were the same as those presented in the familiarization phase, but after hearing a sentence, participants were asked to enter the number of the speaker on the keyboard. Correct responses were followed by a cue tone (“Ding”). If the answer was incorrect, the correct number of the speaker was announced to remind the participant. Each participant heard 5 sentences × 5 speakers × 5 times = 125 trials in total.

Generalization phase (GP)

After practicing, each participant was asked to recognize the voices of 10 novel sentences read by the same 5 speakers as in the practice phase without feedback. Each participant heard 10 sentences × 5 speakers × 1 time = 50 trials in total. Their accuracy in this phase was computed to measure their voice-recognition ability.

Delayed memory phase (DP)

Two weeks later, the participants returned to the lab and performed the same task as in the generalization phase. This retention test allowed us to examine the possible difference in voice memory ability between the blind and the sighted groups.

### 2.3. rs-fMRI Data Acquisition and Preprocessing

The participants were scanned on a SIEMENS MAGNETOM Prisma 3-Tesla magnetic resonance imaging scanner with a 20-channel head coil. High-resolution T1-weighted images were obtained using an MPRAGE sequence (192 slices, slice thickness = 1.00 mm, in-plane resolution = 448 × 512, TR = 2530 ms, TE = 2.98 ms, TI = 1100 ms, flip angle = 7°, field of view = 224 × 256 mm, voxel size = 0.5 × 0.5 × 1 mm^3^). We acquired rs-fMRI data using an echo-planar imaging (EPI) sequence with the following parameters: 64 transversal slices, slice thickness = 2.00 mm, in-plane resolution = 112 × 112, TR = 2000 ms, TE = 30 ms, flip angle = 90°, FoV = 224 × 224 mm, and voxel size = 2 × 2 × 2 mm^3^, 240 functional volumes. During the resting-state scanning, all the participants were blindfolded and instructed to keep their eyes closed and stay awake, but not to think actively about a particular idea as much as possible.

Functional volumes were preprocessed and analyzed using SPM12 (Wellcome Department of Imaging Neuroscience, London, UK) and Data Processing Assistant for Resting-State fMRI pipeline analysis (DPARSF) [49] implemented in MATLAB (MathWorks). The initial 10 functional volumes were discarded to allow for signal stabilization and the subject’s adaptation to the environment. The preprocessing of the remaining 230 volumes included: (1) slice timing correction for acquisition timing differences, (2) realignment of the functional images to correct for head motions and coregistration of functional and anatomical data, (3) regressing out nuisance covariates including Friston 24-head motion parameters [50], white matter signal, cerebrospinal fluid signal, and linear trends, (4) spatially normalizing the realigned images into the Montreal Neurological Institute (MNI) space by using the parameters from the DARTEL algorithm for anatomical images processing [51] and resampled to 2 × 2 × 2 mm^3^, (5) spatial smoothing using a 4 mm FWHM Gaussian kernel, and (6) a band-pass filter (0.01–0.10 Hz) to reduce the effect of low-frequency drift and high-frequency noise.

### 2.4. Seed-Based FC Analysis

To explore the reorganization of the specific FCs between voice- and face-sensitive areas, we performed seed-based FC analyses and compared them across the blind and the sighted groups. As the selectivity of voice recognition is particularly pronounced in the right hemisphere [52,53], our analyses focused on the voice- and face-sensitive regions in the right hemisphere.

Drawing from the outcomes of the earlier research that identified the voice-sensitive regions in the human auditory cortex, known as the ‘temporal voice areas’ [17], we defined two “voice patches” along the right STS/STG as regions-of-interest (ROIs): the right posterior ‘temporal voice areas’ (TVAp, MNI coordinate: x = 42, y = −35, z = 3) and the right anterior ‘temporal voice areas’ (TVAa, MNI coordinate: x = 55, y = −2, z = −7). Considering both temporal and extra-temporal regions play important roles in performing a voice-recognition task, we selected the right amygdala (MNI coordinate: x = 20, y = -8, z = −12) [17] and right inferior frontal gyrus (IFG, to be exact, the posterior triangularis; MNI coordinate: x = 53, y = 26, z = 26) [9], both of which show reliable voice sensitivity. The face-sensitive areas were identified based on a quantitative meta-analysis of fMRI studies on sighted participants [9], including the right FFA (MNI coordinate: x = 41, y = −53, z = −19) and the right OFA (MNI coordinate: x = 40, y = −81, z = −5) as ROIs. We also ran an exploratory analysis of the voice/face-sensitive areas in the left hemisphere (Please refer to Appendix A for detailed descriptions of the ROIs).

In the ROI-to-ROI FC analyses, 6 mm radius spheres were created centering in the coordinates of 6 ROIs, and the time course for each seed was extracted by averaging the time courses of all voxels in the ROI for each participant. Then, the synchrony of the time series between the 6 ROIs was assessed by Pearson’s correlation coefficients, which were transformed into Fisher *z*-scores. Next, we performed two-sample *t*-tests to examine the differences between the transformed correlation coefficients of the two groups. To describe the relationship between the reorganization of FCs and voice-recognition ability, brain–behavior correlational results were obtained with the *p*-values corrected by the False Discovery Rate (FDR) method for multiple comparisons. Comparisons of the correlation coefficients were performed between the two groups according to the method proposed by Diedenhofen & Much (2015) [54].

## 3. Results

### 3.1. Behavioral Results

The behavioral accuracy data in the generalization phase and delayed memory phase indicated the two participant groups’ voice-recognition ability across the different conditions (Figure 2). The group difference was analyzed using a three-way repeated-measures ANOVA with Time (GP, DP) and Language (CN, JP) as within-subject factors, and Group (SC, EB) as a between-subject factor. The ANOVA results revealed a significant main effect of Group (*F* (1, 40) = 5.439, *p* = 0.025, $\eta_{p}^{2}$ = 0.120), indicating that overall, the blind participants performed better than their sighted counterparts. There was also a significant main effect of Language (*F* (1, 40) = 42.472, *p* < 0.001, $\eta_{p}^{2}$ = 0.515) with no Group-by-Language interaction (*F* (1, 40) = 0.280, *p* = 0.600, $\eta_{p}^{2}$ = 0.007), indicating that the two groups were equally more accurate in Chinese voice recognition than in Japanese voice recognition. We also found a significant main effect of Time (*F* (1, 40) = 37.030, *p* < 0.001, $\eta_{p}^{2}$ = 0.481) and a marginally significant Group-by-Time interaction (*F* (1, 40) = 3.711, *p* = 0.061, $\eta_{p}^{2}$ = 0.085), indicating that both groups were more accurate at GP than at DP with a greater difference of accuracy between the repeated tests in the sighted (*F* (1, 40) = 33.698, *p* < 0.001, $\eta_{p}^{2}$ = 0.457) than in the blind group (*F* (1, 40) = 8.255, *p* = 0.006, $\eta_{p}^{2}$ = 0.171). There was no significant three-way (Group by Time by Language) interaction (*F* (1, 40) = 0.595, *p* = 0.445, $\eta_{p}^{2}$ = 0.015).

### 3.2. Changes in Functional Connectivity among the Voice- and Face-Sensitive Areas in the Early Blind

To delineate the alterations in the functional connectivity among voice- and face-sensitive areas in the blind subjects, we compared the average FCs of the six seeds between the sighted and early blind groups using two-sample *t*-tests (Figure 3a,b, Appendix A).

Within the voice-recognition network, the early blind group exhibited significantly higher FCs than the sighted group between the amygdala and following regions: TVAp (*t* (40) = −2.947, *p*-FDR = 0.027, *d* = 0.911), TVAa (*t* (40) = −2.565, *p*-FDR = 0.030, *d* = 0.793), and IFG (*t* (40) = −2.794, *p*-FDR = 0.029, *d* = 0.863) (see Figure 3b,c).

For the FCs between the voice and face recognition networks, we found significant reductions in the strength of FFA-TVAa (*t* (40) = 2.599, *p*-FDR = 0.030, *d* = 0.803) and OFA -TVAa (*t* (40) = 2.720, *p*-FDR = 0.029, *d* = 0.840) in the early blind group compared with the sighted group (Figure 3b,c). In contrast, the FC strengths of FFA-IFG and OFA-IFG were significantly enhanced in the early blind group relative to the sighted group (*t* (40) = −3.214, *p*-FDR = 0.019, *d* = 0.993; *t* (40) = −4.702, *p*-FDR < 0.001, *d* = 1.453; see Figure 3b,c). Similarly, the results of seed-based analyses in the left hemisphere showed the FC strength of L.FFA/OFA-L.IFG was higher in blind individuals relative to sighted individuals (see Appendix A for a full description).

### 3.3. Correlations between Voice-recognition Ability and the Strengths of FC

We conducted Pearson’s correlations between voice-recognition ability and the strength of functional connectivity for each group. The results showed that a stronger connectivity between TVAp and FFA was associated with better voice recognition only in the sighted participants (CN-GP: *r*-SC = 0.640, *p*-FDR = 0.015; *r*-EB = −0.014, *p*-FDR > 0.05; *r*-SC > *r*-EB, *p* = 0.021). On the other hand, in the blind participants, a stronger connectivity between the amygdala and FFA was associated with better voice-recognition performance (CN-GP: *r*-EB = 0.607, *p*-FDR = 0.075; *r*-SC = −0.315, *p*-FDR > 0.05; *r*-EB > *r*-SC, *p* = 0.002; JP-DP: *r*-EB = 0.765, *p*-FDR < 0.001; *r*-SC = 0.395, *p*-FDR > 0.05; *r*-EB > *r*-SC, *p* = 0.077). Refer to Figure 4 and Appendix A for more details.

Although the *p*-values were not significant after being corrected for multiple comparisons of correlation analysis (*p* uncorrected < 0.05, *p*-FDR > 0.05), there was a tendency that the stronger FCs of TVAp/TVAa-amygdala and TVAa-FFA/OFA were associated with better performance for voice recognition only in the sighted group, while the stronger FCs of amygdala-OFA and FFA-OFA associated with better voice-recognition performance only in the early blind group (Figure 4a, Appendix A).

## 4. Discussion

In the present study, we investigated how vision loss shaped the neural substrates for voice recognition by resting-state fMRI in early blind individuals and sighted controls. Behavioral results replicated previous findings on the superiority of voice retention memory [43] and the significant effect of language familiarity on voice recognition in the blind group [44]. ROI-wised functional connectivity analyses evidenced a significant enhancement in the functional coupling between the amygdala and TVAp/TVAa/IFG in the early blind. We also found stronger FCs between FFA/OFA and IFG but weaker FCs between FFA/OFA and TVAa in blind than in sighted participants. Furthermore, we analyzed the correlations between FCs and voice-recognition accuracy in each group. Our results showed that better behavioral performance was associated with stronger FC between TVAp and FFA only in sighted individuals but stronger FC between the amygdala and FFA only in early blind individuals.

### 4.1. Enhanced Internal Connections of Voice Perception Network in the Early Blind

Recognizing a person by voice involves multiple processes. Correspondingly, a large network of distributed brain areas is involved in the processing of voice identity, including not only temporal voice areas as the core parts but also subcortical (such as the amygdala) and prefrontal cortices as the extended regions [15,17,18]. One of the key findings in the present study is that the intranetwork connections among the voice-sensitive areas were enhanced in the blind group, indicating reorganization within the intact voice-recognition system associated with visual impairment. Moreover, our results showed that the alterations in the voice perception network were not confined to TVAs but also included the extended parts of the network, especially the amygdala. The amygdala is involved in the processing of emotional voices in the blind [37]. Some evidence has suggested that the amygdala is also associated with the processing of voice and face traits regardless of the affective characteristics [17,55]. A recent fMRI study in patients with primary visual cortex impairment has confirmed that the amygdala is involved in the processing of socially salient but emotionally neutral facial expressions [56]. In the current study, emotionally neutral stimuli were used, and more accurate (GP) and delayed (DP) performances were associated with a stronger connection between the amygdala and TVA, thus providing further support for the role of the amygdala in speaker identity recognition irrespective of emotional valence.

We also observed that the FC between the amygdala and IFG was enhanced in the blind group. The inferior frontal regions are involved in recognizing learned-familiar persons [9,57], and extensive evidence has indicated that the basolateral complex of the amygdala projects to plenty of regions (e.g., the prefrontal cortex and hippocampus) associated with learning and memory [58,59,60]. The enhanced pathway between the amygdala and IFG observed in this study, therefore, might be a neural basis for enhanced ability to establish and consolidate the link between voice trait and identity in blind people. This result is corroborated by previous evidence that blind individuals learn faster in voice-recognition training [35,42,61] and are more accurate in delayed voice-identity recognition compared with sighted counterparts [43]. Taken together, the strengthened intra-network functional connectivity between the distributed voice-sensitive areas might play a critical role in the voice recognition of the early blind. More specifically, the amygdala appeared to be a key component in the voice perception network.

### 4.2. Reorganization of the Internetwork Connections between the Voice- and Face-Sensitive Areas in the Early Blind

Neuropsychological and neuroimaging studies provide mounting evidence for the multimodal integration of facial and vocal information during identity processing [26,28]. Voice- and face-sensitive areas are functionally and anatomically connected for transferring the identity information during voice recognition [19,22,23,24]. The exchange of information between the two systems facilitates identity processing in sighted people [27,62]. The findings of the current study in the early blind are consistent with previous work by showing a positive association between the FC of TVAp-FFA and voice-recognition performance.

More importantly, we found that the strengths of the FC between FFA/OFA and TVAa were reduced in the blind group, indicating the absence of crossmodal integration of facial and vocal information due to visual deprivation. Similarly, it was reported that auditory deprivation would introduce a significant reduction of fractional anisotropy and increment of radial diffusivity in the V2/V3- and FFA-TVA connections [25]. We speculate that vision loss in blind individuals disrupts the visual input to FFA, leading to the absence of crossmodal sensory integration in the FFA-TVA pathways and the consequent reduced connectivity in the FFA-TVA pathways. Our speculation is further supported by consistent findings across previous studies that the TVA (particularly the anterior part) as an association area receives identity information (such as gender or age) conveyed both by facial and vocal stimuli [24,26,63,64].

Taken together, our result that the functional connectivity between the voice- and face-sensitive areas promoted voice identity processing is consistent with the well-documented “integrative model” of personal recognition in sighted people [27,62,65], but the absence of crossmodal sensory integration induced by visual deprivation leads to reduced coupling between the voice- and face-sensitive areas in early blind individuals.

### 4.3. Neuroplastic Changes of the Face-Sensitive Areas in the Early Blind

The blind group outperformed the sighted group in the delayed memory phase (but not in the generalization phase) during the voice-recognition task. It is possible that early blindness promoted the long-term memory consolidation of speaker identity. Indeed, we found that blind participants’ performance during the delayed memory phase was positively correlated with the FC between the amygdala and FFA. Previous studies have provided strong evidence for the direct white matter pathway [66] and high functional coupling between the amygdala and FFA [67,68]. A meta-analysis study revealed that the superficial subregion nucleus of the amygdala and FFA were primarily involved in cognitive memory [69]. Our data suggest that the efficiency of functional connectivity between the amygdala and FFA may modulate the long-term memory storage for voice through the retained pathways in the early blind.

Moreover, the FC between FFA and OFA was associated with voice-recognition accuracy only in the blind group. Given that the network of face perception was composed of distributed patches such as FFA and OFA [10] and that the FC between FFA and OFA plays a critical role in face perception among sighted people [70], our result indicates that face-sensitive areas can retain their functional selectivity in blind people [30,32,71]. This is consistent with previous findings that the FFA could be activated by auditory-only voice recognition without corresponding face training in sighted people [19,23]. In addition, a recent fMRI study using multivoxel pattern analysis and functional cortical mapping techniques demonstrated that blind individuals could develop category selectivity (face, body, etc.) in the ventral-temporal cortex which was strikingly similar to the sighted controls [72]. However, given that visual impairment disrupts cortical processing of facial properties, it remains inconclusive whether the disrupted face system was dedicated to early or late stages of voice processing or both.

Meanwhile, we observed that FCs between FFA/OFA and IFG were enhanced in the blind group. The inferior frontal areas are considered as extended parts of both the voice perception network [17,18,73] and the face perception network [10]. The frontal regions associated with voice recognition are directly adjacent to the regions involved in face recognition [9]. Further investigations are needed to clarify the precise role of IFG and its subregions in voice perception.

## 5. Conclusions

The clear group differences in the current behavioral and resting-state fMRI data reveal plastic changes in the neural substrates for voice recognition associated with early visual deprivation. Specifically, the internal links of the intact voice system were enhanced, while the connections between the core part of the voice system and the disrupted face system were decreased in the early blind. Despite visual deprivation in blind individuals, intrinsic brain activities independent of experimental tasks showed that the face system was not excluded from the processing of personal identity; instead, it was found to be actively involved in voice recognition via the connections between the core face-sensitive areas (e.g., FFA and OFA) and the amygdala/IFG. These findings are in line with the “metamodal” theory that the two systems conduct similar computational operations during face and voice processing during the functional reorganization [28], which may facilitate blind individuals’ talker identity recognition and their adaptation to the social environment in daily life.

## Figures and Tables

**Figure 1 brainsci-13-00636-f001:**
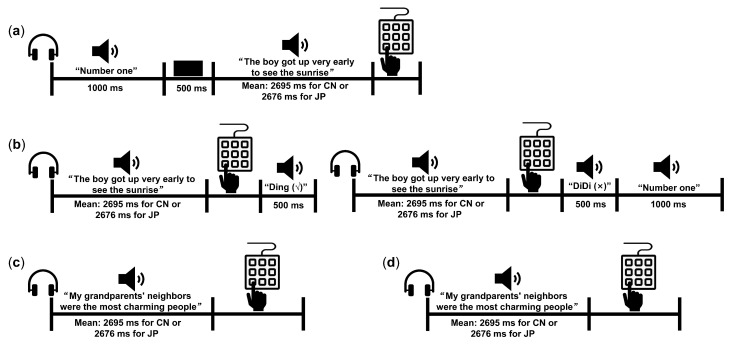
Illustration of behavioral experimental design. (**a**) Familiarization phase: participants heard a number designating the speaker followed by a training sentence read by that speaker. Then participants pressed the key to begin the next trial. (**b**) Practice phase: after hearing a sentence, participants were asked to type in the number of the speaker. Correct responses were followed by a cue tone (“Ding”). Incorrect responses were followed by a cue tone (“DiDi”), then the correct number of the speaker was presented. (**c**) Generalization phase (GP): after hearing a sentence, participants were asked to enter the number of the speaker on the keyboard without feedback. (**d**) Delayed memory phase (DP): the stimuli and procedure were the same as in the generalization phase.

**Figure 2 brainsci-13-00636-f002:**
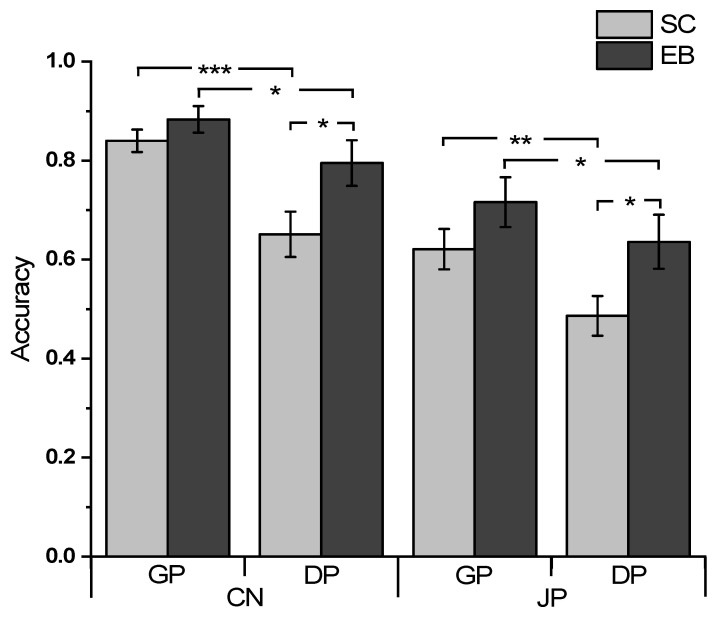
Mean voice-recognition performance of the sighted and early blind participants in each condition. Abbreviations: CN, Chinese condition; JP, Japanese condition; GP, Generalization phase; DP, Delayed memory phase; SC, Sighted control; EB, Early blind. Significance indicated by * (*p* < 0.05), ** (*p* < 0.01), and *** (*p* < 0.001). Error bars represent the standard error of the mean.

**Figure 3 brainsci-13-00636-f003:**
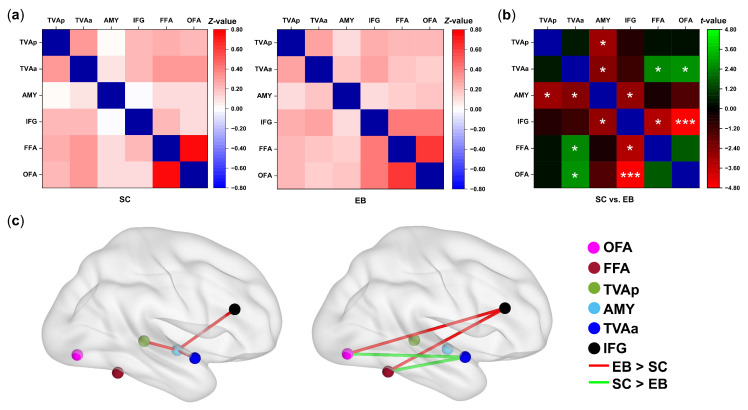
The between-group difference in the seed-based FCs. (**a**) Connectivity matrix reporting average FC between 6 ROIs of the sighted (SC) and blind (EB) participants. The color bar represents the Fisher *z*-scores of FCs. (**b**) The color bar represents the *t*-values for the contrast (SC vs. EB). Significance indicated by * (*p*-FDR < 0.05) and *** (*p*-FDR < 0.001). (**c**) The sagittal views show the significant differences in ROI-to-ROI FCs between the sighted and blind participants. Left panel: Changes of FCs within voice-sensitive areas; Right panel: Changes of FCs between the voice- and face-sensitive areas. Abbreviations: FFA, fusiform face area; OFA, occipital face area; TVAp, posterior “temporal voice areas”; AMY, amygdala; TVAa, anterior “temporal voice areas”; IFG, inferior frontal gyrus. Red lines represent enhanced connectivity in the blind (EB > SC); Green lines represent decreased connectivity in the blind (SC > EB).

**Figure 4 brainsci-13-00636-f004:**
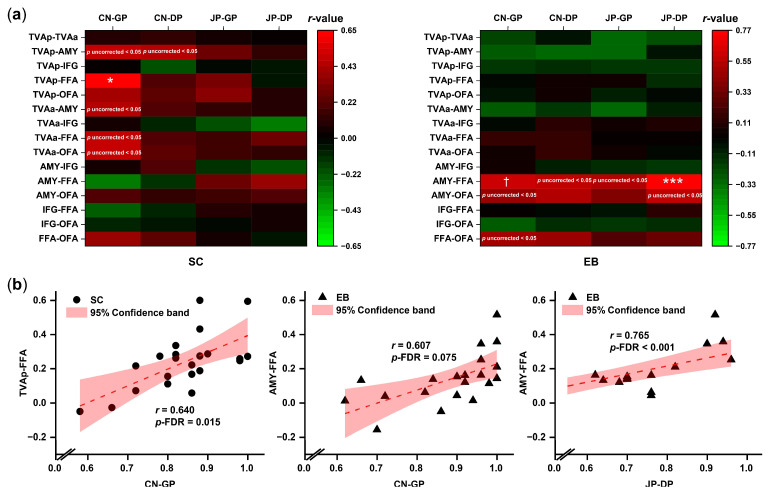
FC-behavior correlations. (**a**) Correlation analyses between the mean FC strength and mean voice-recognition accuracy of the sighted (SC) and blind (EB) participants. Abbreviations: CN-GP, accuracy in the Chinese condition in the Generalization phase; CN-GP, accuracy in the Chinese condition in the Delayed memory phase; JP-GP, accuracy in the Japanese condition in the Generalization phase; JP-GP, accuracy in the Japanese condition in the Delayed memory phase. The color bar represents Pearson’s *r*. Significance indicated by † (*p*-FDR = 0.075), * (*p*-FDR < 0.05) and *** (*p*-FDR < 0.001). (**b**) Scatterplots show the significant correlations (after FDR corrected) between the ROI-to-ROI FCs and the voice-recognition performance accuracy.

## Data Availability

The data that support the findings of this study are available from the corresponding author, W.P., upon reasonable request.

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
