# Peer review of "Visual Deprivation Alters Functional Connectivity of Neural Networks for Voice Recognition: A Resting-State fMRI Study"

_brainsci, 2023, doi:10.3390/brainsci13040636_

Round 1

Reviewer 1 Report

In this study, the authors investigated how visual deprivation shapes the neural underpinnings of voice identity recognition. Previous studies have shown alterations of functional connectivity in congenitally blind individuals, but none has looked at the specifics of voice processing. Additionally, the study provides insight into how vision shapes multisensory networks, because it is known from studies in sighted people that voice- and face-sensitive brain regions directly interact with each other during voice recognition. Thus, this study makes an important contribution to the question of how early visual experience influences the formation of multisensory interactions and of the voice processing network in the human brain.

The authors acquired behavioral and rsfMRI data from a sample of 20 early blind participants and 22 sighted controls matched for age, educational level, and musical experience. Behavioral data were collected using a paradigm that the authors had also used in their previous studies. The results replicate the authors’ previous findings and those of others. In short, overall voice recognition accuracy was higher in early blind at both timepoints and the group difference was more pronounced two weeks after the training.

The exciting part of the study is the analysis of resting state functional connectivity between voice- and face-sensitive areas, and how connectivity relates to voice recognition performance. The authors defined 6 seed regions from the literature (TVAa, TVAp, amygdala, IFG, FFA, OFA) and analyzed their functional connectivity. The results show a reduced functional connectivity between voice-sensitive and face-sensitive areas in blind compared to sighted participants but an enhanced functional coupling between these two regions and the other components of the voice processing network. Further group differences were observed in the correlations between voice recognition accuracy and the strengths of the FCs of the 6 seeds.

The manuscript is very well-written. The introduction gives a comprehensive and concise overview of the relevant literature. The methods and results are very clearly described and sufficiently detailed. The figures are well-prepared, illustrate the main points and help the reader to understand the content of the paper. The conclusions of the authors are valid and comprehensible.

I have only a few minor comments:

-        Visual input during the first months of life has a critical role for brain development. By including 4 early blind subjects in the blind group, this study is less rigorous than the authors’ previous studies which included congenitally blind individuals only. Sample sizes in the rare population of congenitally blind humans typically comprised between 10 and 20 participants in other studies. I wonder why the authors have not restricted the sample to congenitally blind individuals.

-        The last two paragraphs of the Introduction can be shortened by changing the structure and omitting repetitions.

-        I would like the authors to explicitly state whether any of the participants was involved in one of their previous studies. I consider this as relevant for the interpretation of the behavioral results.

-        line 221: The reader might be interested in the results of the left hemisphere as well. I would suggest running an exploratory analysis in the left hemisphere, to report the results in the Supplements and to briefly summarize them in the text.

-        line 223: I had to read the first sentence of this paragraph a few times to understand that the authors selected their seed regions from the literature. The sentence seems to imply that a voice localizer scan was conducted by the authors.

-        line 326: “These FC results are consistent with previous findings on rs-fMRI in the blind [39-41].” The sentence is not needed in the summary of the results and seems to weaken the results of the authors. I would move it to a different section and elaborate the sentence, i.e. describe *what* is consistent with previous findings.

-        Line 383-389: The authors speculate in line 381/382 that “the FFA-TVA pathways are primarily established by cross-modal information exchange instead of a top-down feedback projection” and give some evidence for this in the following lines. Firstly, this seems to very speculative and secondly, it did not become clear to me why/how the study with the prosopagnosic participant and the observation that the TVA receives facial and vocal identity information would support such as claim. I think the authors should leave this part out.

Reviewer 2 Report

This manuscript does a great job in using resting-state functional connectivity in describing the relationship between voice recognition performance and the FCs among voice- and face-sensitive areas in early blind individuals. Here are a few suggestions to help improve the manuscript.

This reviewer is very much impressed with the FC-behavioral correlations shown in Figure 4, however, there does not seem to be statistical comparisons between the correlations between the blind group and the sighted group. This reviewer recommends that a statistical comparison be performed between correlations could be done using a procedure similar to that given at this link

https://stats.stackexchange.com/questions/145514/how-to-compare-two-pearson-correlation-coefficients
